

# Uncovering hidden specific diversity of Andean glassfrogs of the *Centrolene buckleyi* species complex (Anura: Centrolenidae)

Luis Amador[1,2], Andrés Parada[1], Guillermo D'Elía[1] and Juan M. Guayasamin[3,4]

[1] Instituto de Ciencias Ambientales y Evolutivas, Universidad Austral de Chile, Valdivia, Chile
[2] Departamento de Investigación Científica, Universidad Laica Vicente Rocafuerte de Guayaquil, Guayaquil, Ecuador
[3] Instituto BIÓSFERA-USFQ, Laboratorio de Biología Evolutiva, Colegio de Ciencias Biológicas y Ambientales COCIBA, Universidad San Francisco de Quito, Quito, Ecuador
[4] Centro de Investigación de la Biodiversidad y Cambio Climático, Ingeniería en Biodiversidad y Recursos Genéticos, Facultad de Ciencias del Medio Ambiente, Universidad Tecnológica Indoamérica, Quito, Ecuador

## ABSTRACT

The glassfrog *Centrolene buckleyi* has been recognized as a species complex. Herein, using coalescence-based species delimitation methods, we evaluate the specific diversity within this taxon. Four coalescence approaches (generalized mixed Yule coalescents, Bayesian general mixed Yule-coalescent, Poisson tree processes, and Bayesian Poisson tree processes) were consistent with the delimitation results, identifying four lineages within what is currently recognized as *C. buckleyi*. We propose three new candidate species that should be tested with nuclear markers, morphological, and behavioral data. In the meantime, for conservation purposes, candidate species should be considered evolutionary significant units, in light of observed population crashes in the *C. buckleyi* species complex. Finally, our results support the validity of *C. venezuelense*, formerly considered as a subspecies of *C. buckleyi*.

# INTRODUCTION

Species delimitation—the process by which species boundaries are determined—is important and a challenge for characterizing the biota of biodiversity hotspots (*Myers et al., 2000*). Achieving a taxonomic scheme that reflects the evolutionary history of organisms is critical for both theoretical (characterizing biodiversity) and practical (designing conservation strategies) reasons (*Esselstyn et al., 2012*). Although species delimitation ideally uses multiple lines of evidence (*Padial et al., 2010*), DNA sequences play an important role in species-level lineage identification (*Chambers & Hebert, 2016*; *Pentinsaari, Vos & Mutanen, 2017*). Sequences have been recently, and more frequently, analyzed under coalescent-based methods (*Pons et al., 2006*; *Yang &*

Corresponding authors
Luis Amador,
amadoroyola@gmail.com
Juan M. Guayasamin,
jmguayasamin@gmail.com

*Rannala, 2010*; *Camargo et al., 2012*; *Reid & Carstens, 2012*; *Fujisawa & Barraclough, 2013*; *Zhang et al., 2013*). The main goal of coalescent-based species delimitation is to identify evolutionarily independent lineages, where each lineage represents a single species (*Fujita et al., 2012*; for the conceptualization of the species category, in the so called Generalized Lineage Concept, see *De Queiroz, 1998*, *1999*, *2007*). Coalescent-based methods, which allow testing alternative hypotheses on the divergence of a lineage, are expected to reduce the subjective bias introduced by researchers, avoiding using ad hoc thresholds (i.e., degree of morphological, ecological, and/or percentage of sequence divergence) as criteria to establish species limits. As such, these methods have become a common tool for delimiting species, both to propose candidate species as well as to describe new species (*Leaché & Fujita, 2010*; *Páez-Moscoso & Guayasamin, 2012*; *Crivellaro et al., 2018*). However, these sequences-based methods have several assumptions that must be met (see *Carstens et al., 2013*; *Talavera, Dincă & Vila, 2013*; *White et al., 2014*), as well as limitations and drawbacks according to the characteristics of the analyzed data, including erroneous results with species of recent diversification (*Wei et al., 2016*; *Jacobs et al., 2018*), and species diversity overestimation (*Sukumaran & Knowles, 2017*). Therefore, conclusive species delimitation studies must have an integrative approach (*Dayrat, 2005*; *Padial et al., 2010*).

Proposing candidate species, which currently is mostly done on the basis of molecular evidence (*Correa et al., 2017*; *Hurtado & D'Elía, 2018*; *Lin, Stur & Ekrem, 2018*) can guide future taxonomic research, allowing one to direct the costlier efforts (e.g., field collections, morphological assessment of large specimen series) to specific taxonomic and geographic areas of interest. In turn, these efforts result in an acceleration of the discovery and validation of new species (*Dellicour & Flot, 2015*; *Vitecek et al., 2017*), which is relevant in the current era of biodiversity crisis. In addition, candidate species can be considered evolutionary significant units, which in turn can be subject to conservation actions (*Moritz, 1994*).

The taxon *Centrolene buckleyi* (*Boulenger, 1882*) has a large distribution inhabiting montane primary and secondary forests in high tropical Andean zones (1,900–3,300 msnm), as well as inter-Andean scrubland and Páramo environments of Colombia, Ecuador, and northern Peru (*Duellman & Wild, 1993*; *Guayasamin et al., 2006*; *Rada & Guayasamin, 2008*; *Guayasamin & Funk, 2009*) (Fig. 1). The glassfrog *C. buckleyi* has a relatively simple taxonomic history; only two taxonomic forms, *venezuelense* (*Rivero, 1968*) and *johnelsi* (*Cochran & Goin, 1970*), are associated to it. The later was synonymized under *C. buckleyi* by *Ruiz-Carranza & Lynch (1991)*. Meanwhile, the form *venezuelense* was considered as a subspecies of *C. buckleyi* until it was elevated to species level as *C. venezuelense* by *Myers & Donnelly (1997)*, based on the argument that it is highly unlikely that the distribution of *C. buckleyi*, with its type locality in Ecuador, reaches Venezuela. *Señaris & Ayarzagüena (2005)* agree in considering *venezuelense* at the species level, given that it presents morphological and acoustic differences with respect to typical *buckleyi*. *Cisneros-Heredia & McDiarmid (2007)*, based on morphological characters, stated that further research is needed to evaluate the distinction of *venezuelense* from *buckleyi*. Even after the removal of *venezuelense*, distinct lines of

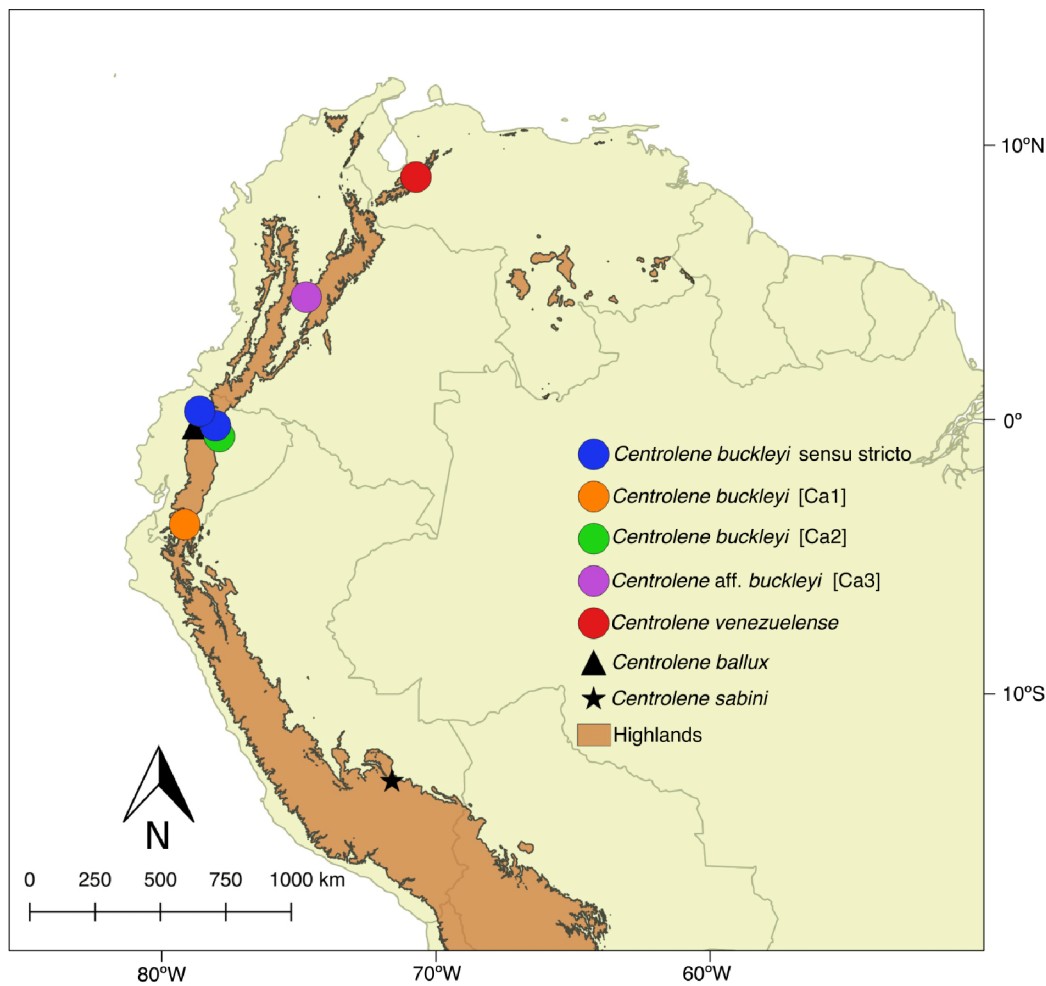

**Figure 1 Distribution of *Centrolene buckleyi* species complex. *Centrolene buckleyi* sensu stricto (KU 178031 Imbabura province, Ecuador and MZUTI 763 Napo province, Ecuador, blue circles); *C. buckleyi* [Ca1] Shucos, Zamora province, Ecuador (orange circle); *C. buckleyi* [Ca2] Yanayacu, Napo province, Ecuador (green circle); *C. aff. buckleyi* [Ca3] MAR 371 Chingaza National Park, Cundinamarca, Colombia (purple circle); *C. venezuelense* EBRG 5244 Páramo de Maraisa, Mérida, Venezuela (red circle). *C. ballux* in Ecuador (black triangle) and *C. sabini* in Peru (black star).**

evidence suggest that as currently understood, *C. buckleyi* may encompass more than one species. For instance, Colombian populations are regarded as *C. aff. buckleyi* as a way to denote uncertainties in their identity (*Guayasamin et al., 2008*). Moreover, *Guayasamin et al. (2006)* identified significant call variation among populations of *C. buckleyi*; similarly, several studies have shown the lack of monophyly for this taxon (*Guayasamin et al., 2006, 2008*; *Guayasamin & Funk, 2009*; *Pyron & Wiens, 2011*; *Castroviejo-Fisher et al., 2014*; *Twomey, Delia & Castroviejo-Fisher, 2014*). As such, in current catalogues (*Guayasamin et al., 2017*) *C. buckleyi* is regarded as a species complex.

Given these antecedents, herein, using an expanded molecular dataset, and phylogenetic and coalescent-based species delimitation analyses, we evaluate the species limits of
populations currently allocated to the taxon *C. buckleyi*. Our study identifies four distinctive lineages, three of which should be considered as candidate species.

## MATERIALS AND METHODS

A total of 34 sequences of the 12S and 16S mitochondrial genes of the genus *Centrolene* were analyzed; of these, 10 belong to the *C. buckleyi* species complex. The outgroup was formed with sequences of *Nymphargus*, a genus closely related to *Centrolene* (*Guayasamin et al., 2008*; *Castroviejo-Fisher et al., 2014*; *Twomey, Delia & Castroviejo-Fisher, 2014*). As such, the analyzed matrix totaled 47 sequences (see Table S1). Sequences were downloaded from Genbank or gathered by us (Table 1 and Table S1). The matrix includes sequences of the *C. buckleyi* species complex from four localities throughout its distributions in the eastern Andean foothills of Ecuador (Table 1; Fig. 1). Specimen collection was done under research permit (MAE-DNB-CM-2015-0017) issued by the Ministerio del Ambiente of Ecuador (MAE). Additionally, morphological characters of all specimens were examined by one of the authors (JMG).

### DNA extraction and sequencing

For newly generated sequences, genomic DNA was extracted from 96% ethanol-preserved muscle tissue samples using a modified salt precipitation method based on the Puregene DNA purification kit (Gentra Systems, Minneapolis, MN, USA). We amplified two mitochondrial genes 12S and 16S using the primers t-Phe-frog and t-Val-frog developed by *Wiens et al. (2005)*. PCR reactions follow the protocol described by *Guayasamin et al. (2008)*. Cycle sequencing reactions were performed by Macrogen Labs (Macrogen Inc., Seoul, Korea). All fragments were sequenced in both forward and reverse directions with amplification primers. Sequences were deposited in GenBank (Table 1; Table S1).

### Phylogenetic analyses

Four matrices were created; a single matrix with 12S, a single matrix with 16S, concatenation of both fragments (these three matrices included all sequences *Centrolene* + outgroup), and one matrix formed only by sequences of *Centrolene* (34 terminals concatenated with 12S–16S mtDNA genes). Sequences were aligned in MAFFT v7 under an automatic strategy (*Katoh & Standley, 2013*). The aligned matrix was imported into Aliview (*Larsson, 2014*), where segments that presented ambiguous alignments were excluded. All positions containing only gaps were deleted. The best nucleotide substitution model was selected with ModelFinder (*Kalyaanamoorthy et al., 2017*) using the Bayesian information criterion and was the same for the first three data sets with all sequences of *Centrolene* + *Nymphargus* (TIM2 + I + G). For the matrix with only sequences belonging to *Centrolene* the selected best model was TIM2 + R3. This last data set was used to infer the input genealogy in species delimitation analysis. The first three were used to conduct phylogenetic analyses.

Phylogenetic trees were obtained using Maximum Likelihood (ML) and Bayesian inference (BI). ML trees were inferred in IQ-TREE (*Nguyen et al., 2015*); nodal support was assessed with 1,000 ultrafast bootstrap replicates (*Minh, Nguyen & Von Haeseler, 2013*).

**Table 1 Names, museum codes, localities, and GenBank accession numbers of sequences of specimens of *Centrolene buckleyi* species complex analyzed in this study.**

| Specie/Taxon | Museum code | GenBank number 12S | GenBank number 16S | Latitude | Longitude | Elevation msnm | Locality | Country/ Province or State |
|---|---|---|---|---|---|---|---|---|
| *Centrolene buckleyi* s.s | KU 178031 | EU663338 | EU662979 | 0.3025 | −78.6186 | 3010 | cerca a Lago Cuicocha | Ecuador. Imbabura |
| ***Centrolene buckleyi* s.s** | MZUTI 763 | MH844843 | **MH844849** | −0.2189 | −78.0444 | 3012 | Zona de humedal en camino Oyacachi-El Chaco | Ecuador. Napo |
| ***Centrolene buckleyi* [Ca1]** | MRy 547 | MH844838 | **MH844844** | −3.8193 | −79.1592 | 2633–2923 | Shucos | Ecuador. Zamora Chinchipe |
| ***Centrolene buckleyi* [Ca1]** | Mry 548 | MH844839 | **MH844845** | −3.8193 | −79.1592 | 2633–2923 | Shucos | Ecuador. Zamora Chinchipe |
| ***Centrolene buckleyi* [Ca2]** | MZUTI 83 | MH844840 | **MH844846** | −0.6133 | −77.8974 | 2187–2190 | Yanayacu Biological Station | Ecuador. Napo |
| ***Centrolene buckleyi* [Ca2]** | MZUTI 84 | MH844841 | **MH844847** | −0.6133 | −77.8974 | 2187–2190 | Yanayacu Biological Station | Ecuador. Napo |
| ***Centrolene buckleyi* [Ca2]** | MZUTI 85 | MH844842 | **MH844848** | −0.6133 | −77.8974 | 2187–2190 | Yanayacu Biological Station | Ecuador. Napo |
| *Centrolene* aff. *buckleyi* [Ca3] | MAR 371 | EU663339 | EU662980 | 4.4660 | −74.7333 | 3035 | Sitio Monte Redondo. P.N. Chingaza | Colombia. Cundinamarca |
| *Centrolene venezuelense* | EBRG 5244 | EU663359 | EU663000 | 8.8419 | −70.7311 | 2450 | Páramo de Maraisa | Venezuela. Mérida |
| *Centrolene venezuelense* | MHNLS 16497 | EU663360 | EU663001 | 8.7092 | −70.9822 | 2100–3050 | Cordillera de Mérida | Venezuela. Mérida |

**Note:**
Sequences generated in this study are in bold.

BI analyses were conducted with MrBayes 3.2 (*Ronquist et al., 2012*) using two parallel runs of four Markov chains that were allowed to run for ten million generations and that were sampled every 1,000 generations. The first 25% of the sampled trees were discarded as a burnin, prior to constructing a consensus tree with the remaining sample. Phylogenetic trees were visualized using FigTree 1.4.3 (http://tree.bio.ed.ac.uk/software/figtree/). Clades with ML values (BV) equal or above 75% and posterior probabilities values (PP) equal or greater than 0.95 were considered as strongly supported. MEGA 7.0 (*Kumar, Stecher & Tamura, 2016*) was used to estimate genetic distances between sequences of the 16S gene with a bootstrap procedure of 1,000 replicates.

## Methods for delimiting species

We used two coalescence-based methods, and their Bayesian implementation, to delimit species on the basis of variation of 12S and 16S sequences.

### Generalized mixed Yule coalescent and Bayesian general mixed Yule-coalescent

The generalized mixed Yule coalescent (GMYC) method uses ML statistics and takes an estimated ultrametric and bifurcating genealogy from a single-locus as input (*Pons et al., 2006*; *Fujisawa & Barraclough, 2013*). The time calibrated-ultrametric tree was obtained with BEAST 2 (*Bouckaert et al., 2014*) using the temporal calibration scheme

outlined by *Castroviejo-Fisher et al. (2014)* for the most recent common ancestor of *Centrolene* and *Nymphargus*. We conducted two-independent analyses to check for consistency in the results under a relaxed clock model and a birth–death model of speciation. Each analysis was run for 20 million generations logging every 1,000 generations. BEAST log files were checked for convergence and for ESS values above 200 using Tracer v1.6 (*Rambaut, Suchard & Drummond, 2014*). Maximum clade credibility tree was estimated with TreeAnnotator v2 (distributed as part of BEAST) with the sampled trees after discarding the first 25% as burn-in. Outgroups were removed with *drop.tip.simmap* function of R v. 3.3.2 (*R Core Team, 2016*) package phytools (*Revell, 2012*). The GMYC method attempts to model the transition point between cladogenesis (Yule process) and the population level process of allelic coalescence, using the assumption that cladogenesis will occur at a much lower rate than coalescence (*Carstens et al., 2013*; *Tang et al., 2014*). GMYC was fitted to the ultrametric gene tree to delimit the species boundaries using single (GMYCs, *Pons et al., 2006*) and multiple threshold models (GMYCm, *Monaghan et al., 2009*). We compared the likelihood of single and multiple transition model with likelihood of null model via a likelihood ratio test. These analyses were performed with the package Splits (*Ezard, Fujisawa & Barraclough, 2009*) in R, after removing zero-length branches and making the tree fully dichotomous. We also performed a Bayesian general mixed Yule-coalescent (bGMYC) analysis (*Reid & Carstens, 2012*), which takes into account the uncertainties in the estimation of the genealogy. The analysis was done with the R package bGMYC (*Reid & Carstens, 2012*) in R, which calculates the posterior marginal probabilities of species boundaries. This was performed with a post-burn-in sample of 100 trees sampled from the posterior distribution of trees. For the bGMYC analysis, the priors of parameters $t1$ and $t2$ were set at 4 and 100, respectively. The bGMYC analysis was performed with 50,000 generations, with a burnin of 10%, and a thinning interval of 1,000 samples.

### Poisson tree processes and Bayesian Poisson tree processes methods

This method models speciation and coalescence events in terms of numbers of substitutions (*Zhang et al., 2013*). Poisson tree processes (PTP) provides hypothesis of species delimitation based on a gene tree (not necessarily ultrametric), using heuristic algorithms to identify the most likely classification of branches in processes at the level of populations and species (*Tang et al., 2014*). We also used Bayesian Poisson tree processes (bPTP), which is the Bayesian implementation and updated version of the PTP method. Moreover, the result of the search for the maximum probability in PTP is part of the results of bPTP. This implementation produces Bayesian posterior probability values (PPV) of delimited species using as input the phylogenetic tree (the same as in PTP). A higher Bayesian value (PPV >0.90) at one node indicates that all descendants of that node are more likely to belong to the same species (*Zhang et al., 2013*). The PTP and bPTP analyses were performed on the web server http://species.h-its.org/ptp/.

To discern among incongruent results of the species delimitation analyses, we followed the reasoning of *Carstens et al. (2013)*, relying on those delimitations schemes that are recovered in the majority of the analyses.

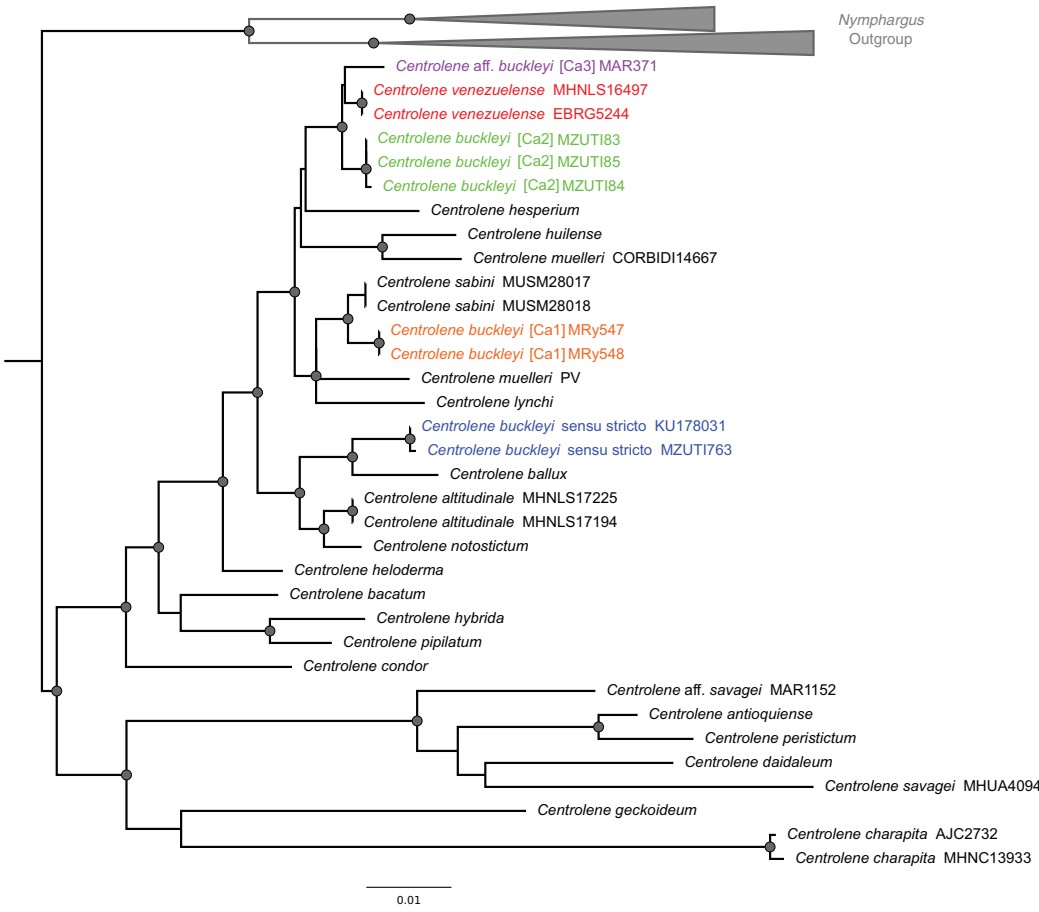

**Figure 2 ML tree depicting phylogenetic relationships of species of the genus *Centrolene* based on the concatenated dataset of 12S + 16S sequences.** Gray nodes represent Bayesian posterior probabilities equal or greater than 0.95 and ML bootstrap values equal or greater than 75%. Species names with different colors represent lineages within the *Centrolene buckleyi* species complex.

## RESULTS

### Phylogenetic relationships

The alignment of 47 nucleotide sequences, including the outgroups, resulted in a total of 1,641 positions in the final dataset. All new sequences were deposited in GenBank. The methods of phylogenetic reconstruction (ML and BI) inferred identical evolutionary relationships, in particular regarding the lineages of the *C. buckleyi* species complex (Fig. 2).

The *C. buckleyi* species complex is not recovered as monophyletic; sequences recovered from specimens of *C. buckleyi* form four main lineages, namely *C. buckleyi* sensu stricto, *C. buckleyi* [Ca1], *C. buckleyi* [Ca2], and *C. aff. buckleyi* [Ca3] (Fig. 2). Specimens of the first lineage come from the proximities of the type locality of *C. buckleyi* (Intag, Imbabura province, Ecuador), as such, hereafter we refer to the first lineage as *C. buckleyi* sensu stricto. This form is sister to *C. ballux* in a strongly supported clade (BV = 99%, PP = 1); *C. buckleyi* [Ca1] is sister to *C. sabini* (BV = 84%,

PP = 0.98), while *C. buckleyi* [Ca2] is sister to a clade formed by *C. venezuelense* and *C.* aff. *buckleyi* [Ca3] from Colombia in a highly supported clade (BV = 96%, PP = 1) (Fig. 2).

## Molecular species delimitation

The results obtained with the GMYCs approach, delimited 26 putative species of the matrix of 34 sequences of *Centrolene*; recognizing *C. buckleyi* sensu stricto, *C. venezuelense*, *C.* aff. *buckleyi* [Ca3], *C. buckleyi* [Ca1], and *C. buckleyi* [Ca2] as different species (Fig. 3, see also Fig. S1). The two specimens of *C. buckleyi* sensu stricto (KU17803, Cuicocha Lake, Imbabura province, and MZUTI763, Oyacachi-El Chaco road, Napo province; distance between locations: about 60 km) were clustered in a single lineage. Similarly, the two specimens of *C. buckleyi* [Ca1] (MRy547 and MRy548, Shucos, Zamora Chinchipe province) and the three individuals of *C. buckleyi* [Ca2] (MZUTI83–MZUTI85, Yanayacu, Napo province) were clustered in a single lineage, respectively. Furthermore, the two specimens of *C. venezuelense* were grouped in a single lineage as well as *C.* aff. *buckleyi* [Ca3] was recovered as a different candidate species. Meanwhile the GMYCm approach found 21 species, GMYCm was the only one of the species delimitation methods that yielded distinct delimitation results, greatly departing from the results of the other analyses (Fig. S2). For instance, GMYCm separated the three specimens of *C. buckleyi* [Ca2] into two different species, while consolidating *C. buckleyi* sensu stricto and *C. ballux* as a single species; similarly, *C. venezuelense* and *C.* aff *buckleyi* [Ca3] that were recovered as a single entity. For both methods (GMYCs and GMYCm), the result of Likelihood ratio test was not significant (LRtest$_{SINGLE}$ = 0.063, LRtest$_{MULTIPLE}$ = 0.058) and the likelihood value of the GMYC model was always higher at both methods, single and multiple (ML$_{SINGLE}$ = 202.026, ML$_{MULTIPLE}$ = 202.117) than the value of the likelihood of null model (L = 199.268). The Bayesian implementation of the method (bGMYC) also delimited 26 putative species, the same that were recovered with GMYCs (Fig. 3, see also the Klee diagram in Fig. S3).

The ML implementation of PTP and the Bayesian implementation of the method (bPTP), considered the topology recovered with MrBayes as a guide tree. These methods delimited 26 putative species with good support, recovering the same delimitation results obtained with GMYC and bGMYC (Fig. 3 and Fig. S4). Further details of the results of the analysis of species delimitation are in Supplementary Information.

Average genetic distances of the 16S matrix, within and between candidate species pairs of the *C. buckleyi* species complex are presented in Table S2. The maximum values between candidate species were observed for the comparison between *C. buckleyi* [Ca1] with *C. buckleyi* sensu stricto and *C. buckleyi* [Ca2] (1.6% and 1.8%, respectively); while the lower values correspond to the comparisons between *C. venezuelense* with *C. buckleyi* [Ca2] and *C.* aff. *buckleyi* [Ca3] (0.7% and 0.4%, respectively). It is also worth highlighting the low values recovered between *C. buckleyi* sensu stricto with its sister taxa *C. ballux* (0.6%) and *C. buckleyi* [Ca1] with its sister taxa *C. sabini* (0.4%).

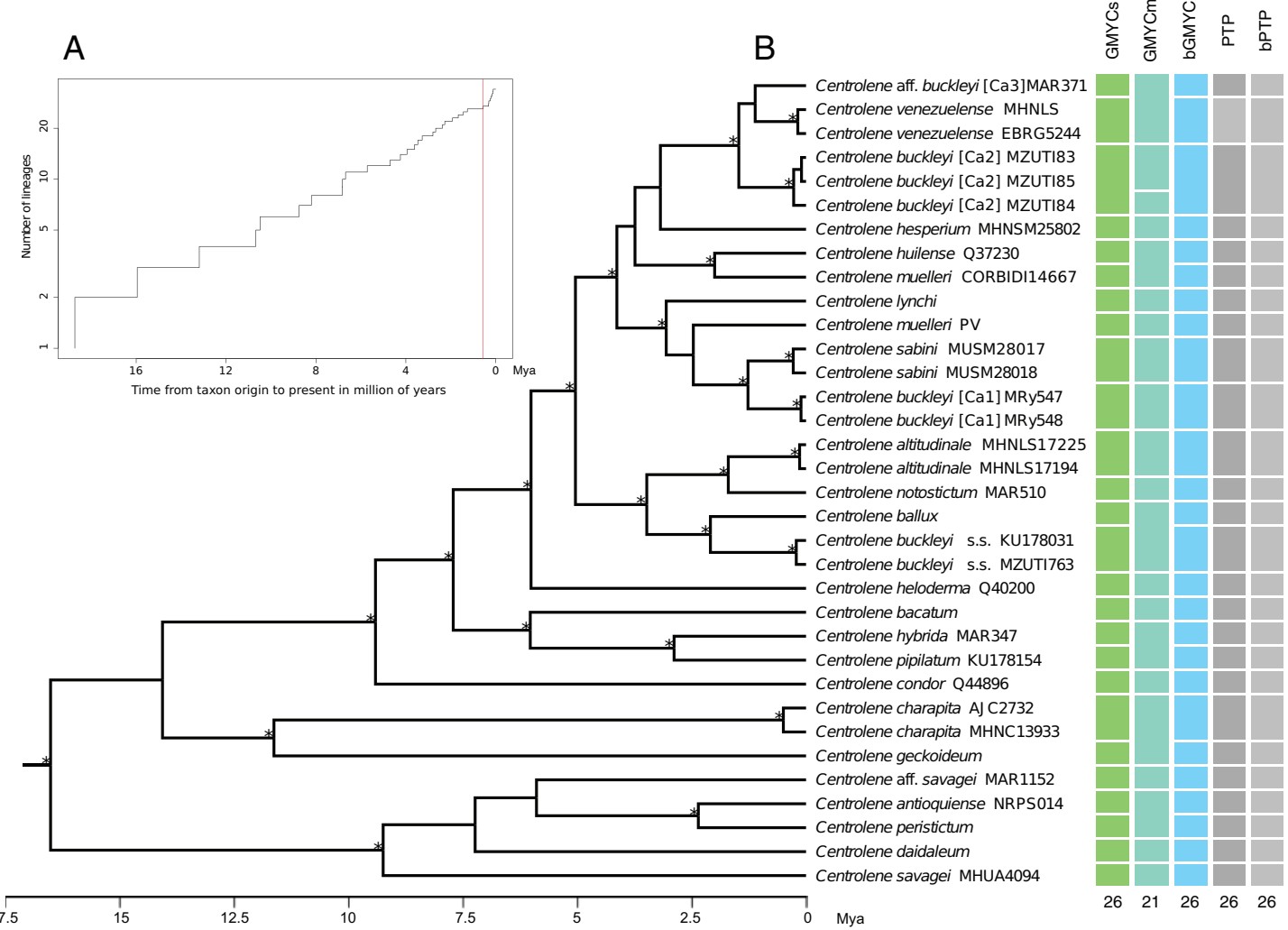

**Figure 3 Maximum clade credibility of DNA sequences of glassfrogs of the genus _Centrolene_ based on mitochondrial DNA (12S and 16S) showing a comparison of the results of distinct species delimitation methods (see text for details).** (A) Figure corresponds to log lineages through time plot, which shows an increase in the rates of branching to the present, probably corresponds to the change from interspecific to intraspecific branching events. (B) The calibrated gene tree is from a BEAST 2 analysis under a relaxed clock with a Birth–Death model tree prior. Node height was determined using mean height across the posterior distribution. Asterisks above branches represent Bayesian posterior probabilities equal or greater than 0.95. Each colored bar represents a candidate species delimited by each method employed. The outgroup (_Nymphargus_) was removed prior to the analyses.

## DISCUSSION

_Centrolene buckleyi_ has been suggested to represent a species complex, requiring therefore, a taxonomic revision (_Guayasamin et al., 2006_; _Cisneros-Heredia & McDiarmid, 2007_; _Hutter, Guayasamin & Wiens, 2013_; _Castroviejo-Fisher et al., 2014_), which in turn would have direct consequence on conservation status and strategies (the _C. buckleyi_ species complex is currently listed as Vulnerable by IUCN; _Guayasamin, 2010_).

In this study, we find four distinct lineages within the _buckleyi_ species complex and corroborate the distinction of _C. venezuelense_ (but see the results of the GMYCm analysis, Fig. 3). Within the current taxonomic definition of _C. buckleyi_, we propose

three candidate species (namely *C. buckleyi* [Ca1], *C. buckleyi* [Ca2], and *C.* aff. *buckleyi* [Ca3]) in addition to the typical form (*C. buckleyi* sensu stricto). This scenario is supported by phylogenetic (ML, BI) and coalescence-based species delimitation analyses (GMYC, PTP). The ML and BI analyses resulted in a tree with topology similar to those of previous studies on glassfrogs (*Hutter, Guayasamin & Wiens, 2013*; *Castroviejo-Fisher et al., 2014*; *Delia, Bravo-Valencia & Warkentin, 2017*), but with an increased sampling within the *C. buckleyi* species complex. In all phylogenetic analyses performed in this study, *C. buckleyi* is not recovered as monophyletic; specimens allocated to this taxon fall into four main lineages: *C. buckleyi* sensu stricto, *C. buckleyi* [Ca1], *C. buckleyi* [Ca2], and *C.* aff. *buckleyi* [Ca3], which are not closely related (except for *C. buckleyi* [Ca2] and *C.* aff. *buckleyi* [Ca3], which are recovered in the same clade with *C. venezuelense* in ML and BI analysis). These lineages, instead, are recovered as sister to distinct glassfrog species (*C. buckleyi* sensu stricto—*C. ballux*, *C. buckleyi* [Ca1]—*C. sabini*, and *C.* aff. *buckleyi* [Ca3]—*C. venezuelense*, respectively). Therefore, the topologies obtained here suggests that there are at least four species in the *C. buckleyi* species complex. We acknowledge, however, that the observed lack of monophyly of the *C. buckleyi* species complex at the mitochondrial genome may be a case of differences between gene and species trees, which may be caused by distinct biological processes, such as mitochondrial DNA introgression. This process has been suggested for cases of other amphibians such as the Nearctic treefrogs of the genus *Dryophytes* (*Bryson et al., 2010*, *2014*) and the toads of the genus *Rhinella* (*Sequeira et al., 2011*). However, without nuclear DNA data we cannot test if the mitochondrial-based tree inferred for the *C. buckleyi* species complex departs from the species tree; as such, we have no reasons to reject these data and assume they represent the true evolutionary history of the group.

Coalescent-based species delimitation methods provide hypotheses for the delimitation of species based on gene trees inferred from DNA sequences (*Fujita et al., 2012*). Previous studies have reported the congruence in the results of methods such as GMYC and PTP when defining putative species (*Lang, Bocksberger & Stech, 2015*; *Thormann et al., 2016*; *Conte-Grand et al., 2017*). Although GMYC and PTP differ in assumptions (e.g., GMYC uses an ultrametric tree and in PTP it is not required), we obtained similar results with both methods. In fact, most of the used species delimitation methods agree in considering *C. buckleyi* as a species complex composed of at least four independently evolving lineages. GMYC provided consistent diversity estimates for BEAST trees (*Talavera, Dincă & Vila, 2013*; *Tang et al., 2014*). This method gives better results when the guide tree is well supported; otherwise, it may tend to overestimate (or underestimate) the number of candidate species (*Fujisawa & Barraclough, 2013*; *Leavitt, Moreau & Lumbsch, 2015*). In addition, GMYC is generally stable in the presence of a certain number of singletons, as is our case (*Monaghan et al., 2009*; *Pons et al., 2006*). The results of GMYC with multiple threshold (which searches for more than one probable scenario of speciation) gave a smaller number of candidate species that the other methods; when the GMYC method was used with a single threshold (GMYCs), results were in line with those of bGMYC, PTP, and bPTP. It should be noted that bGMYC has been successfully applied in other studies that examined species delimitation in

amphibians (*Lawson et al., 2015*; *De Andrade et al., 2016*); however, this method could also fail due to errors associated with unilocus data (e.g., selection, error in gene tree estimation, incomplete lineage sorting; *Satler, Carstens & Hedin, 2013*). Both PTP approaches, ML-PTP, and bPTP (adding Bayesian PPV) gave exactly the same result; these methods simultaneously inferring speciation events based on change in the number of substitutions in the internal nodes (*Zhang et al., 2013*). When visualizing the likelihood plot of each delimitation method, we observed that the MCMC chains converged, which suggests that the PTP results are reliable (Fig. S5). In bPTP, high values of Bayesian support were obtained, which were calculated as the number of occurrences of all the descendants under a particular node, and are the PP that these taxa form a single species under the PTP model (*Zhang et al., 2013*).

The results obtained with the methods GMYC and PTP are consistent; however, this scheme should be viewed with caution, mainly due to intrinsic factors of the mitochondrial genes (e.g., smaller genome size, high mutation rates), to the sampling coverage of the taxa, and to the time of divergence between taxa (*Talavera, Dincă & Vila, 2013*; *Luo et al., 2018*), which could regard intraspecific structure as distinct species (*Satler, Carstens & Hedin, 2013*; *Sukumaran & Knowles, 2017*). For instance, there are results (not within the *C. buckleyi* species complex) that are clearly erroneous, such as the union into a single lineage of two morphologically distant species (*C. charapita* and *C. geckoideum*) with GMYCm method. Therefore, it is also vital to evaluate these molecular species delimitation methods in light of other sources of data (e.g., morphological distinctiveness).

One of the main limitations of our study is that our genetic sampling covers only a fraction of the historical distribution range of the *C. buckleyi* species complex. The limited sampling is consequence of population crashes observed across distributional range of this species complex. For instance, *Bustamante, Ron & Coloma (2005)* mentioned that *C. buckleyi* has disappeared or is rarely found at localities where it used to be abundant. Similarly, *Guayasamin et al. (2006)* carried out intensively fieldwork in Yanayacu during 3 years and only found three individuals of what we recognize as *C. buckleyi* [Ca2]. Finally, historically *C. buckleyi* has been reported in several localities and different vegetation formations in 10 different provinces of Ecuador; however, only four populations had been recorded between the years 1997 and 2007 (*Cisneros-Heredia & McDiarmid, 2007*). Two of these populations correspond to the candidate species that we propose herein (*C. buckleyi* [Ca1] and *C. buckleyi* [Ca2]). Thus, in the scenario of widespread population declines, our findings of distinct independently evolving lineages within the *C. buckleyi* species complex make a strong case for increasing efforts aimed to avoid their disappearance.

Here, we show the existence of candidate species in the *C. buckleyi* species complex. Although the morphology of all candidate species matches the description of *C. buckleyi*, calls from the so far studied candidate species are different (see *Guayasamin et al., 2006*). This observation matches the fact that calls of populations from the Cordillera Occidental of Colombia, approximately 180 km west of the locality of *C. aff. buckleyi* [Ca3] (*Bolívar, Grant & Osorio, 1999*) and those of *C. buckleyi* [Ca2]

(Yanayacu Biological Station (YBS) in northeast of Ecuador) are different. *Guayasamin et al. (2006)* found that the call of *C. buckleyi* [Ca2] in YBS consisted of one to five notes and fundamental frequency = 4,139 Hz which is considerably distinct from that of specimens from Colombia that consists of a single note and fundamental frequency = 5,200 Hz.

It is worth noting that, in some cases, genetic differentiation between species pairs is not related with geographic distance. For example, despite the large geographic gap separating the analyzed populations of *C. buckleyi* [Ca2] with those of *C. venezuelense* and *C.* aff. *buckleyi* [Ca3] (approximately 1,300 and 700 km, respectively; see Table 1), the genetic differences for the 16S matrix are low (0.7% and 0.9%). The same pattern is seen for *C. sabini* and *C. buckleyi* [Ca1] that have a very low genetic distance (0.4%) and approximately 1,300 km of geographic distance. On the other hand, for the pair *C. buckleyi* sensu stricto and *C. buckleyi* [Ca2], whose populations in the cloud forests of the Ecuadorian Napo province, are separated by only 45 km, present a large genetic distance than the previous comparisons (1.4%). Moreover, in general, sister species of *Centrolene* show low levels of genetic divergence. For example, between *C. altitudinale* and *C. notostictum* there is a divergence of 0.7%, a similar low value (0.6%) is found between *C. huilense* and *C. muelleri* (CORBIDI 14667) (see Table S2), these two last sister species were recovered as a single species by the GMYCm method. It has been suggested that most speciation events of *Centrolene* occurred during the last 5 million years, mostly mediated by the Andes uplift (*Lynch & Duellman, 1997*; *Hutter, Guayasamin & Wiens, 2013*; *Castroviejo-Fisher et al., 2014*). This rapid and recent speciation could explain both the low genetic differences found among species, as well as the little morphological divergence observed in the *C. buckleyi* species complex. A similar pattern has been observed in the plump toad *Osornophryne bufoniformis*, another high-Andean anuran species distributed in the northern Andes of Ecuador and Colombia, where highland species also exhibit shallow genetic differentiation (*Páez-Moscoso & Guayasamin, 2012*).

Some of the candidate species that are suggested in our study (Fig. 3) seem to be separated by well-characterized biogeographic barriers. For example, *C. sabini* (Kosñipata valley, Cusco Department, Peru) and *C. buckleyi* [Ca1] (Zamora Chinchipe, southern Ecuador) are separated by the Huancabamba Depression, an important geographic barrier delimiting distinct communities of high-Andean amphibians (see *Catenazzi et al., 2012*; *Hutter, Guayasamin & Wiens, 2013*; *Castroviejo-Fisher et al., 2014*; *Hutter, Lambert & Wiens, 2017*). Another well-supported example of sister species of Andean frogs separated by geographical barriers are rainfrogs of genus *Pristimantis*, such as *P. cedros* and *P. pahuma* separated by the Guayllabamba River in northern Ecuador (*Hutter & Guayasamin, 2015*).

The inferences drawn from this study should be taken as conservative when evaluating species boundaries of the *C. buckleyi* species complex, mainly because our population sampling is relatively low and to the use of single-locus based methods. The hypothesis posed here should be used as a preliminary perspective of species boundaries and not as the only evidence necessary to circumscribe species (*Leavitt, Moreau & Lumbsch, 2015*).

However, the lack of monophyly of *C. buckleyi* as currently delimited, constitutes strong evidence of the existence of hidden specific diversity. Having said that, before formalizing any taxonomic change (i.e., describing and naming any new species), further studies integrating morphological variation, as well more geographical samples, additional behavioral (calls) and genetic data (nuclear markers), are needed to test our taxonomic hypothesis (*Olave, Solà & Knowles, 2014*; *Sukumaran & Knowles, 2017*).

## CONCLUSIONS

We highlight that our study is in line with several others showing a pattern of high levels of cryptic diversity in amphibians of tropical South America, including glassfrogs (*Páez-Vacas, Coloma & Santos, 2010*; *Funk, Caminer & Ron, 2012*; *Hutter & Guayasamin, 2012, 2015*; *Gehara et al., 2014*; *Twomey, Delia & Castroviejo-Fisher, 2014*; *Ortega-Andrade et al., 2015*; *Tarvin et al., 2017*). As such, we note that, in this era of biodiversity crisis, it is urgent to increase the rate in which the biodiversity is characterized, taking advantage of new and traditional tools but, mainly, by facilitating research, an issue that requires the collaboration of both scientists and governmental authorities that regulate research activities.

## ACKNOWLEDGEMENTS

We thank César Marín, Natalí Hurtado, Ewan Twomey, two anonymous reviewers, and the editors for their valuable comments on an earlier version of this contribution. LA wants to thank Mayra García, Dharma Amador, and Gael Amador for their constant support. Thanks to José Núñez for his advice on analysis of coalescence-based species delimitation.

### Funding

This work was supported by the Secretaría Nacional de Educación Superior, Ciencia, Tecnología e Innovación (SENESCYT), Comisión Nacional de Investigación Científica y Tecnológica (CONICYT). Guillermo D'Elía's research is supported by Fondo Nacional de Desarrollo Científico y Tecnológico FONDECYT (No. 1180366). Juan M. Guayasamin's research is supported by Universidad San Francisco de Quito (Collaboration Grants 5521, 5467, 5447, 11164, Fondos COCIBA, and Fondos Semillas Biosfera-USFQ). The funders had no role in study design, data collection and analysis, decision to publish, or preparation of the manuscript.

### Grant Disclosures

The following grant information was disclosed by the authors:
Secretaría Nacional de Educación Superior, Ciencia, Tecnología e Innovación (SENESCYT).
Comisión Nacional de Investigación Científica y Tecnológica (CONICYT).
Fondo Nacional de Desarrollo Científico y Tecnológico FONDECYT (No. 1180366).

Universidad San Francisco de Quito (Collaboration Grants 5521, 5467, 5447, 11164, Fondos COCIBA, and Fondos Semillas Biosfera-USFQ).

## Competing Interests

The authors declare that they have no competing interests.

## Author Contributions

- Luis Amador conceived and designed the experiments, performed the experiments, analyzed the data, contributed reagents/materials/analysis tools, prepared figures and/or tables, authored or reviewed drafts of the paper, approved the final draft.
- Andrés Parada analyzed the data, contributed reagents/materials/analysis tools, authored or reviewed drafts of the paper, approved the final draft.
- Guillermo D'Elía contributed reagents/materials/analysis tools, authored or reviewed drafts of the paper, approved the final draft.
- Juan M. Guayasamin conceived and designed the experiments, performed the experiments, contributed reagents/materials/analysis tools, authored or reviewed drafts of the paper, approved the final draft.

## Field Study Permissions

The following information was supplied relating to field study approvals (i.e., approving body and any reference numbers):

Collected specimens were approved by the Ministerio del Ambiente of Ecuador (MAE) provided research permits (MAE-DNB-CM-2015-0017).

## DNA Deposition

The following information was supplied regarding the deposition of DNA sequences:

The 12S–16S sequences of *Centrolene buckleyi* generated in this study are available in GenBank (codes MH844838–MH844849)

## Data Availability

The raw data is provided in the Supplemental Files.

## Supplemental Information

Supplemental information for this article can be found online at http://dx.doi.org/10.7717/peerj.5856#supplemental-information.

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
