# Peer review of "Uncovering hidden specific diversity of Andean glassfrogs of the Centrolene buckleyi species complex (Anura: Centrolenidae)"

_PeerJ, doi:10.7717/peerj.5856_

## Round 0.1 · original submission · Major Revisions

As you'll notice below, there have been substantial differences between reviewers in their assessment of your manuscript. Please pay particular attention to the comments of reviewers 1 and 3 and make sure you properly address all of the raised issues.

Reviewer 1 ·

Basic reporting

I think the manuscript is written in proper English. The references are adequate, but there are some minor errors in the citations and references (one missing). The structure, figures and tables seem good to me. However, I have my concerns about the relevance of the results (detailed in the comments to the authors).

Experimental design

The work is within the aims and scope of the journal, but I think that it fails to fill a gap of knowledge (see comments to the authors).

Validity of the findings

See comments to the authors.

Additional comments

This study aimed to evaluate the species diversity of populations allocated to Centrolene buckleyi and the taxonomic status of C. venezuelense, for which a series of standard unilocus species delimitation methods were applied. As a result, three candidate species were identified and the distinction of C. venezuelense was ratified. The methodological approach and the interpretation of the analyses are relatively simple so there should not be ambiguities in the taxonomic decisions derived from the results. However, I have several serious disagreements with this study.

1. The authors first highlight the advantages of using delimitation methods based on coalescence (lines 40-46), but finally recognize that these methods actually diagnose genetic structure, not species (lines 278-279). Consequently, the authors consider their results as "a first assessment of species boundaries of the C. buckleyi species complex" and recognize that their single-locus data provide "a preliminary view of species boundaries" (lines 275-278). So, what is the contribution of this research to the taxonomy of the C. buckleyi complex? One of the arguments that suggest that C. buckleyi is a complex is "its lack of monophyly", which can be inferred from some of the articles cited (e.g. Guayasamin 2008, Twomey et al. 2014). Therefore, the only novelty of this study is that C. buckleyi would be composed of at least four lineages that could correspond to different species. I agree "that in this era of biodiversity crisis, it is urgent to increase the rate in which the biodiversity is characterized" (lines 292-293), but it is not clear to me why the authors did not formalize any of their findings. I suppose that the authors are working on the description of one or more of the candidate species, but by not including this final step, there is practically no progress in solving the problem: C. buckleyi is still a species complex (something that could already be inferred from previous analyses) and C. venezuelense would be a good species (something that other authors had said).

2. The distribution of C. buckleyi complex covers more than 1000 km and the authors only included specimens from five localities. Is this enough to evaluate the diversity in a taxon that apparently includes several species whose distributions are obviously not known? Why were not more populations assigned to C. buckleyi included? From Ecuador, for example? (Cisneros-Heredia & McDiarmid 2007) I think that sampling is insufficient, in number of localities and specimens, to solve a taxonomic problem of that magnitude.

3. My third main point is about the interpretation of the results of the delimitation and distances analyses. Usually, the idea behind using various methods of delimitation is to seek instances of congruence between them (i.e. candidate species supported by all of them) and to resolve cases of inconsistency by going to information from other sources, unless the authors favor one method over another. Here I do not see clearly what strategy the authors applied. First, the methods of delimitation recognized between 21 and 25 species, but where is explained what criteria were used to choose between the methods? In the discussion, apparently, the authors favor the delimitation obtained with bGMYC because of its greater sensitivity and because it has been used successfully in other studies of amphibians (and you will always find examples where any method works well!). However, the fact is that there are many examples in the literature where this method fails in the delimitation of species, mainly due to overesplitting (e.g. Miralles & Vences 2013, Hamilton et al. 2014, Pentinsaari et al. 2016), and the same could be applied to other methods. Conversely, the results of PTP analyses are discarded because they group C. sabini and C. buckleyi 2 as a single evolutionary unit. But why is the conspecifity of C. sabini and C. buckleyi 2 an equivocal delimitation? (line 254) Can not there be a species of amphibian that is distributed along 1300 km? In this case, the short branch lengths that separate these taxa are striking (Fig. 2), so intermediate samples are required to elucidate this issue. Second, the two methods try to accommodate intra- and interspecific processes along the branches of a phylogeny so the question is, how does the present sampling influence the performance of the methods, when most of the data are one specimen per species? Of course the proponents of the method say that it works very well under a wide spectrum of conditions (e.g. Fujisawa & Barraclough 2013), but the present data set do not allow to evaluate if the method is really distinguishing intra/interspecific patterns of molecular variation. And third, regarding the genetic distances, what is the point of calculating the distances only between the focal populations? This does not add additional information to what one can observe in the phylogeny, but it does provide a clue that the authors neglect. Are the genetic distances that were obtained between the candidate species low or high compared to the other species in the group? Although the authors do not provide all the distances between species, it seems that the values between C. buckleyi 3 with C. aff. buckleyi and C. venezuelense are the lowest of all the species of the genus included. I am aware that genetic distances are not an unequivocal indication of species status, but those values are well below the values that have been proposed as a thumb rule for the limit of intra/interspecific divergence (3-5%) for the 16S in amphibians (e.g. Vences et al. 2005, Fouquet et al. 2007, Matsui et al. 2016, Walker et al. 2018). According to this yardstick, even the other candidate species (and surely other species included in the phylogenetic analysis) do not reach enough divergence to be considered as full species (I think that the pair C. sabini-C. buckleyi 3 fall in this category). I reiterate, there is no fixed value to define the boundary between the intra/interspecific divergence, but the genetic distances between some populations of C. buckleyi and other closely related species are quite low enough to consider the possibility that they are conspecific.

4. My last main point is about the other types of evidence that the authors provide as additional support for the distinction between candidate species. In the discussion they point out that glassfrog species in general are altitudinally segregated, giving the example of C. buckleyi 1 and C. buckleyi 3. However, the data of C. buckleyi 3 are only from the Yanayacu Biological Station and of C. buckleyi 1 only from two localities (Table 1). What about all the other populations of C. buckleyi from Ecuador? (Cisneros-Heredia & McDiarmid 2007) Is there enough geographic data to claim that both species segregate altitudinally? On the other hand, the authors cite differences in calls between both species, but the data are from one specimen per population. Similarly, is there enough data to establish if there are significant differences in this trait? My opinion is that the knowledge of the phylogeny, biogeography and bioacoustics of this group of anurans is still too incomplete to support those taxonomic suggestions with such fragmentary data.

In short, I believe that this work does not represent a significant advance in the taxonomy of the C. buckleyi complex because its main result is only to confirm that this taxon is a complex, formed by at least four species. I think that sampling is insufficient to solve such a complex problem and that the authors do not adequately argue their methodological choices. Moreover, It is not clear to me why authors provide only partial information to support the specific status of some of the candidate species (and why they refrain of describing them formally), and in turn ignore their own genetic distance data, which suggest that some of the populations may be conspecific.

I have a few minor comments, but I did them in the corresponding text in the pdf. Also, I detected several minor errors, including spelling, and other formatting errors that I also indicated in the pdf.

Annotated reviews are not available for download in order to protect the identity of reviewers who chose to remain anonymous.

Reviewer 2 ·

Basic reporting

The article is well written and the conclusions supported by the analyses they performed. The authors proposed some taxonomic changes for a group of glass frogs and they are aware of the limitations of their data set regarding sampling numbers. However, the authors do not mention the weaknesses of working with mtDNA only, and that their results might be biased because of this. More about this in the section 'Validity of the findings'

I have some comments for specific points in the manuscript:

L143 change model to models
L150 higher? how much is higher?

Please use a constant term throughout the manuscript : "a single molecular entity" or "a single evolutionary unit"

L200 C. aff. buckeyi in italics

L255 What do you mean by "intrinsic factors of the 12S and 16S genes"

L261 (0,6% y 0,8% respectively : be consistent in the decimal separator, use period instead of comma, same with thousand separator, use comma instead of period

L277 In my opinion, there is a more fundamental problem than the analytical outcome of the coalescent methods when using one locus. The main problem is the incomplete history resulting from analysing only mtDNA loci.

L300 Finantial change to Financial

Figure 1 It would be useful to know where the samples of the closely related species were taken.

Figure 2 Bayesian and ML totally concordant? Why did you pick this tree? Please also explain in the text.

Figure, please be consistent on how C. buckleyi 1 is labeled in the figures, you refer to those sample at least in 3 different ways

Table 1 there is no column for the Genbank accession numbers

What is the difference between Fig 2 and Figure S2? Is Figure S2 really needed?

Experimental design

Please provide more details about how the BEAST analysis was performed. I assumed that you calculated relative divergence rates only, but this is not explained in the manuscript.

Table S2. how were these distances calculated?

Validity of the findings

One interesting outcome is the paraphyly of the study species Centrolene buckleyi, a result previously found by other studies and confirmed with the newly analysed samples in this manuscript. The authors describe this pattern but do not provide possible explanations for it. For example, is this the result of past introgression and mtDNA sequence capture? This phenomenon has been observed in other frogs e.g. Bryson et al 2010). Although the authors lack nuclear data to test for this hypothesis, at least this possibility should be mentioned and considered as several studies suggest this might be a widespread phenomenon. Additionally, unless I missed it, the authors mentioned that to improve the species delimitation it would be necessary to add more geographical samples, how about adding more markers, specifically nuclear markers?
The authors did not mention anything about differences in morphology among the samples studied, nor possible differences between the C. buckleyi groups and the species they resulted related to. It would be useful to provide information about this. How the specimens were identified? Is there a possibility that they have been misidentified?

How about geographic distribution changes? The authors mentioned that some of the taxa that now have similar mtDN sequences are rather far away. What is known about distribution changes in this species or in other species in the area?

The discussion will benefit from considering plausible biological explanations for the results in addition to the methodological aspects that the authors discussed.

Reviewer 3 ·

Basic reporting

The manuscript is well-written and clear, with good coverage of relevant literature. I think the introduction could have a bit more context. It is basically paragraph 1 about species delimitation, and paragraph 2 is already the specifics of the taxonomy of the buckleyi group. I think a paragraph to bridge these two paragraphs could be a good addition. Specifically, it would be good to convey the importance of finding candidate species. Most people would not argue against the significance of actually describing species, but here no species are described so I think you need to "defend" why identifying candidate species but not describing them is still important. It could also be worth mentioning drawbacks, if any, with using coalescent-based delimitation methods.

Experimental design

The methods are described in detail, however, as I mentioned in an email to the associate editor, I do not have expertise in coalescent-based species delimitation, so I cannot adequately review the technical aspects of those analyses.

Validity of the findings

The interpretation of the results is generally sound, although in some cases relies on some subjective acceptance of one delimitation method vs. another. I expand on this in point 2 of the general comments section.

Additional comments

1. Overall the main goal of the paper isn't entirely clear. It seems to me that applying these coalescence methods should be a tool towards some greater objective, like describing a species or maybe using the identified taxa to evaluate some biogeographical scenarios. But in this case the application of the method seems like the endgame. The most obvious way this could be rectified would be to actually describe the species here. That would of course entail a lot more work on the manuscript but it would make crystal clear why the authors would use these delimitation methods. I understand the authors want to get more data before doing this so formal descriptions might be unfeasible. Otherwise, this aspect of the manuscript could be improved by better emphasizing in the introduction why this study is important. I touch on this a bit in the basic reporting section.

2. I think that using a battery of methods kind of undermines the idea that these methods are an improvement over traditional taxonomy in that they reduce subjective interpretations. For example, which of the 5 methods is 'correct' with respect to Ce. sabini and Ce. buckleyi 2? The authors argue that because of biogeography it is extremely unlikely these two are conspecific, but in my opinion it is completely possible (glassfrogs have some crazy distributions, for example we found Cochranella erminea something like 900 km north from the type locality, and some of the Amazonian Hyalinobatrachium have similar stories.). On one hand it's good to see how different methods treat the same dataset, but using all these methods comes at the cost of increased subjectivity during interpretation. I don't think there's a good way around this but this could be touched on in the discussion.

3. Figure 3 could be improved by including all the terminals on the tree. For example, it seems like only a single buckleyi 2 and buckleyi 3 is included in that figure, but from the results section it sounds like all the terminals were included in the analysis. I think showing everything is important for that figure because it allows the reader to see whether the methods are oversplitting (ie. splitting what are definitely single species, like two sequences originating from the same locality). Related to this, including multiple terminals from species where multiple sequences are available (e.g. Ce. charapita, Ce. sabini, possibly others) could help evaluate over/under-splitting, which could broaden the scope of the discussion a bit. For example if GMYCs also splits charapita into two species, you'd have an indication that this method seems too liberal in delimiting species.

Specific comments:

Introduction
lines 29-30: "urgent challenge in biodiversity hotspots". Wording is unclear -- do you mean it's an urgent challenge in identifying biodiversity hotspots?

line 31: "...due to theoretical and practical issues." I don't really understand what you mean here. You don't actually say why it's important, you just say that it is because of issues, which is vague.

line 33: hypenate 'species-level'

line 35: delete 'mostly'

line 58: change to "...differences with respect to..."

lines 62-68: based on this, in particular the polyphyly of buckleyi, it seems obvious it is a species complex. The different populations of this "species" are not even very closely related. It seems to me that what is needed is not another study saying that there are multiple species here. They ought to be described and this manuscript seems like a good place for it, assuming the relevant data is available.

Methods
line 75: Wording is a bit awkward, change to "Nymphargus, a genus closely related to Centrolene."

lines 96-97: All positions containing gaps or only gap-only positions? If it is the former, why would you do this? An indel shared between two taxa could be evidence of homology. Same with missing data -- if all taxa but one had data for a given site, why would you delete that?

line 165: On the trees, buckleyi 1 is called "buckleyi sensu stricto". Maybe change it to "buckleyi 1 (sensu stricto)" so you know exactly what buckleyi 1 in the text is referring to.

Discussion
lines 205-206: The distinction is only corroborated if you disregard the results of the PTP analysis.

line 227: change to "regards"

line 265: change to "here we show" or "here we have shown"



Sincerely,
Evan Twomey

---

## Round 0.2 · Minor Revisions

Please pay particular attention to the comments by the reviewers, particularly the methodological issues raised by Reviewer 1 and the suggestions regarding the scope of the paper indicated by Reviewer 2.

Reviewer 2 ·

Basic reporting

No comment

Experimental design

No comment

Validity of the findings

No comment

Additional comments

The article has improved after the first round of revision, the added detail in methods and justification of the study is welcomed. I have a couple of minor comments:

- I still think that the BEAST methodology is incomplete, specifically regarding the calibration/no calibration of the nodes. This is essential to reproduce any analysis.

- I would have liked to see the accession numbers of all the sequences included in the analyses. As it is currently presented, the reader would have to dig in the Genbank to find the specific sequences used.

Please address the following spelling mistakes:
L.63, The later was synonymized under C. buckleyi by Ruiz-Carranza & Lynch (1991). - Change to 'latter'
L73 C. aff. Buckleyi - change to lower case

Reviewer 3 ·

Basic reporting

As this is a re-review I've consolidated all my comments to a single section (General comments for author)

Experimental design

As this is a re-review I've consolidated all my comments to a single section (General comments for author)

Validity of the findings

As this is a re-review I've consolidated all my comments to a single section (General comments for author)

Additional comments

As this is a re-review, I mainly focused on whether my suggestions were addressed appropriately, however, I couldn't help but notice that many of my concerns echoed those of reviewer 1, just that reviewer 1 used a bit stronger wording than me. My main comments were:

1. Clarify the goal of the paper. A paragraph was added to the introduction to address this. However, I mentioned that it would be good to also discuss drawbacks or limitations of these methods, but it doesn't look like that was added. Basically, the authors argue that proposing candidate species can be useful for guiding taxonomic research and conservation actions. While this is true, it doesn't change the fact that formally describing these species is actually a better and more direct way of doing the same thing.

2. "Reduce subjectivity" in terms of selecting some methods over others. It appears the authors dealt with this appropriately, supporting candidate species inferred by most or all methods.

3. Include multiple terminals from species where multiple sequences are available. The authors dealt with this, adding some extra terminals to the analysis that allows them to address whether putative conspecific lineages are being "oversplit" by the analyses (it appears they are not). Although, one method (GMYCm) gives some results that are really absurd (e.g. lumping C. charapita and C. geckoideum). This latter result does not appear to be mentioned anywhere in the manuscript. In my view this is kind of an important point because it shows that these methods are necessarily evaluated in light of an existing taxonomy, which in my view calls into serious question their use for establishing new taxonomic proposals "from scratch".

New comments:
-- on re-read, the Conclusions section seems a bit preachy. On one hand the authors are stressing how important it is to characterize biodiversity but at the same time do not describe the species here, despite there seeming to be plenty of evidence for doing so (lack on monophyly, call differences). Thus, this seems sort of ironic.

-- The references should be checked in detail. There are many inconsistencies (e.g. using title case occassionally for titles, sometimes including the volume number and sometimes not, inconsistent italics on journal names)

- the GPS coordinates on the genbank table messed up, I guess a formatting error with decimals


Overall, the authors did mostly address my comments, in particular including additional terminals in the analysis, which I feel definitely strengthens the paper. Still, I'm not totally sure why the analyses presented in this manuscript would not simply be included in the formal taxonomic description. In other words, assuming the descriptions will eventually be done, I don't see the reason to publish these data in a separate manuscript rather than just including it in the formal descriptions. These analyses would simply be used to support the taxonomic proposal. In the rebuttal letter the authors argue that identifying candidate species using methods like this is an important first step for (eventual) taxonomic changes, but I disagree, as there has been plenty of excellent taxonomy done prior to the advent of coalescent delimitation methods. To some extent these comments echo the comments of reviewer 1.

If indeed these populations of C. buckleyi have crashed and are in urgent need of conservation, this only underscores the need to formally describe these species. I doubt the Colombian and Ecuadorian governments will be designing conservation strategies based on candidate species (I could be wrong). To me this is a better argument for why these species should be described than why the delimitation methods should be used. There is clearly call data available for some of these lineages (as mentioned in the discussion), when combined with the new samples, compelling phylogenetic results (i.e. lack of monophyly, delimitation results), the description of these species should be straightforward.

In any case, these comments mainly focus on "degree of advance", which is something PeerJ is not interested in. As far as scientific validity I have no issues with the manuscript in its current state. I have made some changes to the tracked-changes manuscript, mainly minor grammatical issues. The sheer volume of tracked changes made it difficult to see exactly what was supposed to be written, so I would suggest that someone (either one of the reviewers or the editor) carefully check any final version for minor errors.

Sincerely,
Evan Twomey

---

## Round 0.3 · accepted · Accept

I believe you properly addressed all of the remaining issues raised by the reviewers. Congratulations for this fascinating study!

#